# The Effects of Adding Art Therapy to Ongoing Antidepressant Treatment in Moderate-to-Severe Major Depressive Disorder: A Randomized Controlled Study

**DOI:** 10.3390/ijerph20010091

**Published:** 2022-12-21

**Authors:** Myungjoo Lee, Han Choi, Jiwon Shin, Ho-Suk Suh

**Affiliations:** 1Department of Medicine, Graduate School, Cha University, Seongnam 13488, Republic of Korea; 2Graduate School of Art Therapy, Cha University, Seongnam 13488, Republic of Korea; 3Department of Psychiatry, CHA Gangnam Medical Center, Cha University, Seoul 135913, Republic of Korea

**Keywords:** depression, major depressive disorder, art psychotherapy, combined therapy, antidepressants

## Abstract

This randomized controlled study aimed to investigate the effects of art psychotherapy on moderate-to-severe major depressive disorder (MDD). Forty-two MDD patients were recruited from a psychiatric outpatient clinic in Seoul, the Republic of Korea. Participants were allocated on a randomized, open-label basis to either an experimental group, wherein they were treated with art psychotherapy added to pharmacotherapy, or a control group, wherein they were treated with pharmacotherapy alone. Pre- and post-test measures of the Hamilton Depression Rating Scale, Beck Depression Inventory-II, and remission rates were measured. The results indicate that patients treated with art psychotherapy and ongoing pharmacotherapy showed slightly greater improvement when compared with pharmacotherapy alone in moderate-to-severe MDD. These results suggest that art psychotherapy could be an effective add-on strategy for the treatment of moderate-to-severe MDD. However, a rigorous test would facilitate a better understanding of art psychotherapy as an add-on strategy for MDD treatment.

## 1. Introduction

Major depressive disorder (MDD) is a severe mental illness with a global lifetime prevalence of 2–21% [1]. It is highly prevalent, affecting 300 million people worldwide [2]. Approximately 50% of the 800,000 suicides that occur every year are associated with depressive episodes. There is robust consistency across reports from different organizations, making it clear that depression is a major risk factor for suicidal thoughts and behaviors [3,4,5,6]. Projections for 2030 estimate that depression will be the leading cause of disability globally [7]. However, less than half of patients with depression are receiving adequate treatment [8].

The increased occurrence and undertreatment of depression is concerning, as it leads to considerable functional impairment in multiple domains of life. It has been shown that the level of functional impairment in patients with depression is comparable to or greater than that of patients with other chronic medical conditions, such as Type I and II diabetes, myocardial infarction, and hypertension [9]. The functional decrement attributable to depression can last even after depressive symptoms are remarkably improved [10]; it is a lasting and debilitating outcome that leads to substantial personal, familial, and social burdens. Above all, residually impaired functioning in MDD has been associated with an increased likelihood of relapse [11]. Studies have indicated that 80% of individuals experiencing their first episode of major depression will have one or more future episodes throughout their lives [12] and four episodes on average in their lifetime [13]. Even when achieving remission, patients with lingering symptoms have been shown to relapse more than those who achieve full remission, without any residual symptoms [14]. Approximately 15% of patients have an unremitting course, and 35% recover but experience recurrence [15]. This is of clinical significance because patients classified as having chronic depression are resistant to treatment [16] and have a higher hospitalization rate than those with non-chronic conditions [17].

The treatment of MDD comprises several options, including pharmacotherapy, psychotherapy, or a combination of both. As the etiology of depression is multifactorial, involving environmental factors as well as genetic risk, initial treatment in MDD may vary depending on treatment history, treatment response, patient preference, and other clinical factors [18]. The Agency for Health Care Policy and Research (AHCPR), Veterans Health Administration/Department of Defense (VHA-DOD), and American Psychiatric Association (APA) have supported the development and publication of clinical practice guidelines that provide evidence-based recommendations for depression treatment. To treat moderate-to-severe depression, combined treatment of pharmacotherapy and psychotherapy is suggested, while psychotherapy is recommended for patients with mild-to-moderate depression [19,20,21].

In an attempt to enhance treatment outcomes, psychological intervention has long been advocated as an adjunctive therapy. A systematic review investigating randomized clinical trials (RCTs) reported that psychological interventions combined with antidepressant use showed a higher response and lower dropout rate than pharmacotherapy alone; the addition of psychotherapy resulted in a marked improvement in response and adherence to treatment [18,22,23,24,25]. In a recent study, two network meta-analyses [26,27] found that the effects of adding psychological intervention to antidepressant use exceeded those of pharmacotherapy, suggesting that the combined treatment is optimal in terms of treatment response, remission, and acceptability.

Studies have documented that adjunctive psychotherapy improves specific symptoms during the treatment of MDD. Studies [28] have examined the relative efficacy of psychotherapy and combined medication for depressive symptoms and found that adjunctive therapy was efficacious in treating specific symptoms such as obsessive thoughts, emotional lability, and feeling hopeless and entrapped. Additionally, a meta-analysis focusing on functioning and quality of life in the management of depressive disorder reported that the combination of psychotherapy and antidepressant pharmacotherapy was more effective in both outcomes than each treatment alone, which might be associated with a patient’s symptom amelioration [29]. Furthermore, it has been reported that psychotherapy added to antidepressants (ADs) even benefits patients with treatment-resistant depression (TRD) in terms of remission and response rates, as well as the treatment of depressive symptoms [30].

Art psychotherapy is a form of psychotherapy that integrates creative processes and psychotherapeutic techniques. It uses art expression as a prime mode of communication; it conveys thoughts and feelings that cannot be explained verbally as an aesthetic product [31]. Images produced in art psychotherapy enhance the communication ability of inherent powers [32]. Art psychotherapy encourages individuals to explore personal meaning, recognize difficult emotions, integrate conflicting thoughts, and facilitate interpersonal relationships between individuals and art psychotherapists through the creative process [32,33]. Furthermore, engaging in art activities is reportedly related to increased serotonin activity [34], a higher incidence of emotional stability [35], and decreased depressive feelings [32]. Above all, taking a holistic approach to the treatment of mental illness, the current trend in psychiatry focuses on psychotherapeutic interventions, such as art psychotherapy. The MDD Clinical Practice Guidelines (NICE 2014; SIGN, 2010) list art psychotherapy as a nonpharmacological treatment strategy [36]. Art psychotherapy has been used in various clinical settings and populations, although few studies have explored its use in MDD. An RCT [37] of 6 weeks of art psychotherapy was found to be effective in decreasing depressive symptoms and improving daily functioning. Existing studies have compared the effects of ADs alone versus pharmacotherapy combined with art therapy. Combined therapy has been found to contribute positively to achieving and sustaining stable remission in patients with recurrent depression [38]. Another RCT [8] investigated the effects of 20 weeks of combined therapy in an older population with MDD. Most recently, an RCT study [39] suggested that art psychotherapy added to ADs is more effective at improving depressive symptoms, anxiety symptoms, and interpersonal complications than ADs treatment alone for reducing the overall severity of MDD. Furthermore, the above study [39] suggests that the treatment alliance between art psychotherapists and patients might contribute to better medication adherence.

This study explored the hypothesis that six weekly art psychotherapies added to ongoing antidepressant pharmacotherapy are more effective at reducing depressive symptoms than AD pharmacotherapy alone in moderate-to-severe MDD. The primary objective was to investigate the effects of combined therapy on moderate-to-severe MDD based on changes in the level of depressive symptoms. The secondary objective was to compare the changes in the levels of depressive symptoms between patients with moderate-to-severe MDD.

## 2. Materials and Methods

### 2.1. Design

An RCT was conducted. Participants were randomly assigned to either the experimental or control group. The experimental group received art psychotherapy in addition to ongoing ADs for moderate-to-severe MDD. In contrast, the control group received AD-based pharmacotherapy alone.

### 2.2. Sample Size

The sample size calculation was based on a repeated measures ANOVA (*d* = 0.5; *α* = 0.05), and a power of 95% was calculated using G*Power 3.1.9.2 for Windows [40]. Considering the dropout rate of 20%, the estimated total sample size required for the study was 38.

### 2.3. Sample

As shown in Figure 1, participants were recruited via a sampling frame of adult MDD patients from January 2014 to July 2015 (first phase) and from February 2017 to July 2017 (second phase) in a psychiatric outpatient clinic in the third largest district of Seoul, the Republic of Korea. Inclusion criteria included (a) a clinical diagnosis of MDD, (b) moderate-to-severe symptoms of MDD at the study onset measured by a Hamilton Depression Rating Scale (HDRS) score of above 14 points, and (c) a selective serotonin reuptake inhibitor (SSRIs) AD regimen. Exclusion criteria included (a) chronic conditions of MDD in which depressive symptoms lasted for two years or longer and (b) taking psychotherapy during the study period. To prevent the risks of hazards and threats to internal variables, participants who (a) withdrew consent, (b) participated in less than three sessions of art therapy, (c) failed to maintain SSRIs dosage recorded at the study outset, (d) changed dosage during the study period, or (e) switched to other AD agents or underwent augmentation were terminated from the clinical experiment.

Overall, 47 MDD patients were screened. Forty-five participants satisfied the inclusion criteria and agreed to participate in the study. Participants were randomly assigned to either the experimental or control group using a random number generator (https://www.random.org: accessed on 4 January 2014). Of the 45 participants, three were excluded. The reasons for declining participation were that two participants did not meet the inclusion criteria, and one participant declined to participate. A total of 42 participants were allocated to the experimental and control groups. Finally, 39 participants completed the experiment, while three dropped out. Two participants stopped clinic visits, and one participant withdrew consent.

### 2.4. Intervention

Art psychotherapy was constructed and rationalized with the following evidence and recommendations: The largest open-label and RCT of ADs with MDD, the sequenced treatment alternatives to relieve depression (STAR*D) study [41], recommends a minimum of 8 weeks of medication adherence before switching ADs. In an 8-week time frame, a meta-analysis study [42] examining the effects of the frequency of IPT, CBT, and psychodynamic therapy sessions on MDD reported that 6–8 sessions of art psychotherapy were effective compared to 20 sessions of verbal psychotherapy. Based on this, a 6-week treatment was established.

Participants in the experimental group received 50 min art psychotherapy sessions consisting of a series of structured activities once a week for six weeks. The details of each art psychotherapy session in this study are as follows: In the first session, drawing your feelings was performed, and information on art psychotherapy procedures and processes was explained in advance, forming a positive therapeutic rapport with patients so they could prepare themselves psychologically. In the second session, free drawing was performed, and unconscious thoughts were expressed through free association to recognize specific problem areas. In the third session, self-portraits were employed to explore how ego identity was affected by major depressive episodes. In the fourth session, the interpersonal inventory [43] was performed for participants in order to understand social support and interpersonal relationship problems. In addition, patients’ unique communication methods related to interpersonal relationship problems were explored. In the fifth session, family drawings were performed to gain insight into family relationships that influence depressive conditions. In the last session, landscape drawings were conducted to offer tolerance for depression with a broader scope. Identical structured art activities were maintained throughout the study period.

### 2.5. Assessments

Demographic information, including age, gender, and duration of illness, were collected.

The Hamilton Depression Rating Scale (HDRS) [44] was designed to measure the severity of depressive symptoms and allows for an objective assessment of changes in symptoms. The scale consists of 17 items focused on somatic symptoms, making the instrument sensitive to the treatment effects of pharmacotherapy. Each item is assessed on a scale of 0–4, with a total score range of 0–50 points. A score of 0–6 is deemed “normal”, whereas a score of ≥8 is deemed “depressed”. A score of 8–13 is deemed “mildly depressed”, a score of 14–18 is deemed “moderately depressed”, and a score of 19 or greater is deemed “severely depressed”.

The Beck Depression Inventory-II (BDI-II) [45] is designed to measure the severity of subjective depressive symptoms. The scale consists of 21 items focused on emotional, cognitive, motivational, and physiological domains. Each item is assessed on a scale of 0–3, with a total score range of 0–63 points. A total score of 0–13 is deemed as “minimal”, whereas a score of 14–19 is deemed as “mild depression”. A score of 20–28 indicates “moderate depression” and a score of 29–63 indicates “severe depression”.

The remission rate was defined as follows: an HDRS score ≤7 is an indicator of complete remission [46,47], and a BDI-II score ≤12 is an indicator of complete remission [46].

### 2.6. Procedure

Self-referral methods were used to recruit participants. The poster was posted in the clinic waiting area, and the information brochure explained the clinical trial. Patients who wished to participate in the experiment were either contacted by art psychotherapists or clinicians for the initial screening arrangement. Before entering the experiment, the purpose, procedures, and possible hazards of the experiment were thoroughly explained by the researchers, and written informed consent was obtained from the participants. Art psychotherapy was administered to the experimental group. The pre-test was conducted before the first art psychotherapy session, and the post-test was conducted after the final art therapy session. The art psychotherapist conducted the BDI-II and collected the demographic information. A psychiatric specialist conducted the HDRS. Neither inter-rater reliability nor independent verification were performed. Participants’ data were recorded on a case report form. Comparable data were obtained in the control group. An identical set of procedures and assessments were administered to both groups, except for the implementation of art psychotherapy.

### 2.7. Statistical Analyses

All statistical analyses were performed using SPSS Statistics 22.0 (IBM Corp., Armonk, NY, USA). The demographic data of the participants were analyzed by performing the chi-square and independent sample *t*-tests to compare variances for the groups and differences between groups at baseline. For the HRDS, BDI-II analyses were performed using repeated-measures ANOVA with groups (experimental versus control; moderate MDD versus severe MDD) as the between-group factor and measurement occasion (pre-test versus post-test) as the within-subject factor. The equality of covariance matrices across groups was tested, and the results indicated that the assumption of the equality of covariance matrices was reasonable. Additional analyses were performed using paired sample *t*-tests, which provided further comparisons of the treatment outcomes between the pre-test and post-test phases within groups. A per-protocol (PP) analysis was performed, including only participants who completed the experiment.

## 3. Results

### 3.1. Demographic Characteristics between the Experimental and Control Groups

The demographic characteristics of the experimental and control groups are presented in Table 1. Depression severities were determined by HDRS.

### 3.2. Primary Outcomes: Experimental versus Control

Box’s M test was conducted to test the equality of the covariance matrices. The equality of covariance matrices of the HDRS (*M* = 4.518, *p* = 0.259) and BDI-II (*M* = 1.409, *p* = 0.743) was verified. The interaction effect time (pre-test versus post-test) × group (experimental versus control) of HDRS and BDI-II was confirmed. Significant differences were observed between the two groups in terms of the differences between the pre- and post-test measurements of the HDRS (*F* = 8.663, *p* = 0.008, η*p*^2^ = 0.302) and BDI-II (*F* = 8.291, *p* = 0.010, η*p*^2^ = 0.304). The differences between the pre- and post-test measurements of the HDRS and BDI-II in the experimental group were significantly higher than those in the control group.

A paired-sample *t*-test was used to verify the changes in the pre- and post-test variables within each group. The results are presented in Table 2. HDRS scores declined significantly in both groups. The BDI-II score declined significantly in the experimental group, whereas there were no differences in the control group.

### 3.3. Secondary Outcomes: Moderate MDD versus Severe MDD

Box’s M test was conducted to test the equality of the covariance matrices. The equality of the covariance matrices of the HDRS (*M* = 6.648, *p* = 0.115) was verified. However, equality of the covariance BDI-II was not verified (*p* < 0.05). The interaction effects time (pre versus post) X group (moderate MDD versus severe MDD) of HDRS were confirmed. Significant differences were observed between the two groups in terms of the differences between the pre- and post-test measurements of HDRS (*F* = 6.852, *p* = 0.016, η*_p_*^2^ = 0.255). The differences between the pre- and post-test measurements of the HDRS in the severe MDD group were significantly higher than those in the moderate MDD group.

A paired-sample *t*-test was used to verify the changes in the pre- and post-test variables within each group. The results are presented in Table 3. HDRS scores declined significantly in both groups.

## 4. Discussion

These results indicate that AD pharmacotherapy alone and six weeks of art psychotherapy added to the ongoing pharmacotherapy in adult MDD were both significantly effective at reducing depressive symptoms. Significantly greater improvements in BDI-II and HDRS in combined therapy at the endpoint were achieved compared with the patients treated with pharmacotherapy alone. In addition, we found that the magnitude of the effects was larger in the add-on treatment than in the standalone pharmacotherapy for improving depressive symptoms. The results, including the effect size shown in this study, are consistent with the findings of existing studies [8,39], in which adherence to pharmacotherapy and art psychotherapy produced significantly better results than pharmacotherapy alone. Furthermore, this finding is also in line with the results of a series of studies [18,26,27,28,29,48,49] that reported that add-on therapy or combined therapy was more effective at improving depressive symptoms than AD pharmacotherapy alone. However, due to the limited number of studies on art psychotherapy, an in-depth cross-analysis was difficult.

This finding is also in line with the results of a series of studies [18,26,27,28,29,48,49] that reported that psychotherapy or combined therapy has clinically significant advantages over pharmacotherapy alone for treating depressive disorders. A meta-analysis of studies comparing psychotherapy with pill placebo [18] and a placebo-controlled, randomized trial that tested the relative efficacy of antidepressant medications (ADM) and cognitive therapy (CT) [48] showed that psychotherapy had an effect size comparable to that of ADM and outperformed placebo. A network meta-analysis that examined the relative effects of psychotherapies, ADs, and combined therapy for treating adult depression [26] showed that the combined treatment was more effective than psychotherapy or ADM alone at achieving a response at the end of treatment. In addition, a study examining the relative efficacy of psychotherapy versus add-on therapy on individual depressive symptoms [27] showed that add-on therapy outperformed psychotherapy for treating some depressive symptoms, but not others. Furthermore, a review study [29] that evaluated the effects of combined therapy and ADM on reducing relapse and recurrence showed that concurrent therapy has a potential advantage for treating more severe depression. These trials compared the relative effects of psychotherapies, ADM, and combined therapy with a representative sample of patients with mild-to-moderate [28] and moderate-to-severe depression [48,49].

According to the results of the subgroup analysis between moderate depression and severe depression, art psychotherapy in the context of ongoing pharmacotherapy was equally effective at reducing depressive symptoms. In terms of symptomatic severity, the head-to-head comparisons of the subjective and objective depressive symptom responses at the endpoint showed significant improvements in both moderate and severe depression. Furthermore, we found a larger effect size for severe than for moderate depression, supporting a greater improvement in depressive symptoms. Our finding is in line with the results of a series of studies [48,49] indicating that add-on or combined treatment is more effective at improving severe depression than AD pharmacotherapy alone. Although these results indicated [48,49] the effects of combined treatment on severe and moderate depression, in-depth cross analyses remain difficult due to the limited number of studies.

Specifically, among the 39 participants who remained in treatment until the endpoint of HDRS, 10.5% and 4.8% of those in the combined therapy and stand-alone pharmacotherapy groups achieved remission, respectively. As for the BDI-II, 5.3% and none of participants in the combined therapy and stand-alone pharmacotherapy groups achieved remission, respectively. In terms of symptomatic severity, remission was evenly scattered across the two severity levels. This result is a markedly lower remission rate than that reported in previous studies [26,27] that used various depression assessment instruments below a specific cut-off to assess remission rates. On the contrary, the above studies [26,27] found relative risk (RR) ratios for the remission rate of 22.0% (RR = 1.22, 95% CI: 1.08–1.39) in the combined therapy, and that it was more effective than stand-alone pharmacotherapy. Although the remission rate is an important predictor of treatment outcome, the remission rate in the present study was very low to claim clinical meaning.

To interpret the results of the experiment, we used per-protocol analysis. Six participants failed to maintain art psychotherapy and were excluded because of threats to internal variables. Specifically, the participants maintained a pharmacotherapy regime. However, they failed to maintain art psychotherapy, and all received a single art psychotherapy session. The exact reasons for the failure to maintain art psychotherapy could not be identified. A possible explanation for the failure to maintain treatment is that six continuous weekly in-person visits to the clinic may have caused treatment burden. This conjecture could be verified with an intention-to-treat (ITT) statistical analysis, which provides a realistic way to view a treatment regimen.

This study had some limitations. First, a more rigorous study design should be used to increase the validity of the results. Although statistical analyses strongly support the benefits of combined therapy, it is difficult to rule out selection bias due to a lack of blinding. Future studies should consider placing an independent blind assessment or a placebo group to achieve better reliability of the study findings. Second, participants with chronic depressive symptoms were excluded. However, it would be worthwhile for future research to target chronic or treatment-resistant depression. Third, the present study evaluated the positive effect, as art psychotherapy has been shown to have only small cases of potentially harmful effects [50,51]. To gain a better understanding of the realistic treatment effects, future research should closely examine the possible negative effects of add-on treatments. Lastly, pre-treatment personal preferences of psychotherapy modalities should be accounted for to increase treatment compliance and adherence.

## 5. Conclusions

This study examined the efficacy of a 6-week combined therapy for the treatment of patients with MDD and pharmacotherapy. The primary outcomes indicated that art psychotherapy combined with pharmacotherapy significantly decreased depressive symptoms in patients who received the combined treatment compared with those who received pharmacotherapy alone. These results are congruent with those of several previous studies that reported the efficacy of combined treatment with art therapy and pharmacotherapy. The secondary outcomes of the present study indicate that patients treated with art psychotherapy with ongoing pharmacotherapy showed a slightly greater improvement when compared to pharmacotherapy alone in moderate-to-severe MDD. More rigorous research is needed to facilitate a better understanding of art psychotherapy as an add-on strategy for treatment aimed at improving depressive symptoms.

## Figures and Tables

**Figure 1 ijerph-20-00091-f001:**
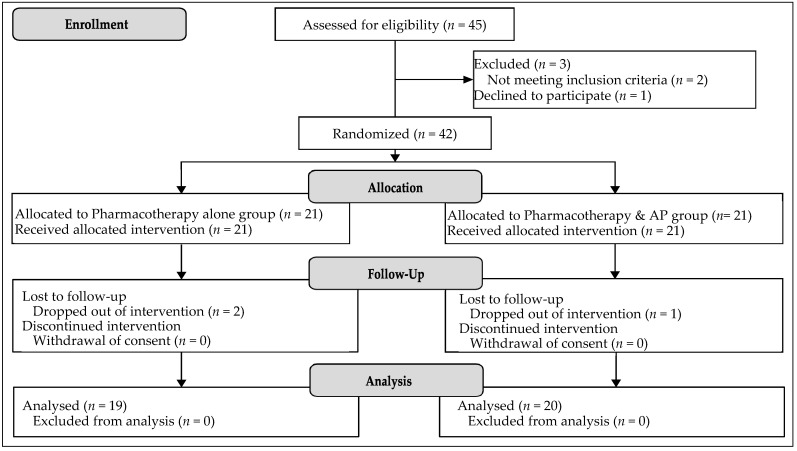
CONSORT flow diagram.

**Table 1 ijerph-20-00091-t001:** Demographic characteristics of participants.

Variables	Sub-Variables	M (SD)		*χ*^2^ or *t*	*p*
		Experimental Group	Control Group		
Age		36.92 (12.4)	40.62 (14.02)	−0.695	0.494
Gender	Male	7 (36.8%)	6 (30.0%)	0.205	0.650
	Female	12 (63.2%)	14 (92.3%)		
Depression Severity (HDRS)	Moderate	10 (52.6)	8 (40.0)	0.626	0.429
	Severe	9 (47.4)	12 (60.0)		
Duration of illness	Less than a month	3 (15.8%)	2 (10.0%)	0.680	0.711
	Less than a year	9 (47.4%)	12 (60.0%)		
	Between 1 and 2 years	7 (36.8%)	6 (30.0%)		

**Table 2 ijerph-20-00091-t002:** Comparison of measurements between the experimental and control groups in moderate-to-severe MDD.

Variables	Group	M (SD)		*t*	*p*	*d* ^3^
		Pre	Post			
HDRS ^1^	Experimental	21.58 (5.09)	15.50 (5.00)	6.304 ***	0.001	1.81
	Control	25.70 (7.29)	23.60 (7.65)	2.272 *	0.049	0.72
BDI-II ^2^	Experimental	29.75 (5.96)	22.92 (5.11)	5.556 ***	0.001	1.60
	Control	33.33 (6.91)	31.56 (6.93)	1.497	0.173	0.50

Abbreviations: ^1^ Hamilton Depression Rating Scale, ^2^ Beck Depression Inventory-II, ^3^ Cohen’s *d*, * Significant at *p* < 0.05, *** Significant at *p* < 0.001.

**Table 3 ijerph-20-00091-t003:** Comparison of measurements between pre- and post-test within the moderate and severe MDD groups.

Variables	Experimental Group	*n*	M (SD)		*t*	*p*	*d* ^3^
			Pre	Post			
HDRS ^1^	Moderate	10	15.45 (1.75)	11.91 (2.59)	4.485 ***	0.001	1.35
	Severe	9	25.09 (4.04)	18.18 (3.79)	6.820 ***	0.001	2.06
BDI-II ^2^	Moderate	11	24.00 (2.54)	20.70 (5.29)	2.810 *	0.020	0.89
	Severe	8	35.64 (7.75)	28.55 (8.44)	5.722 ***	0.001	1.73

Abbreviations: ^1^ Hamilton Depression Rating Scale, ^2^ Beck Depression Inventory-II, ^3^ Cohen’s *d*, * Significant at *p* < 0.05, *** Significant at *p* < 0.001.

## Data Availability

The data supporting the findings of this study are available from the corresponding author upon reasonable request.

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
