# Peer review of "The Effects of Adding Art Therapy to Ongoing Antidepressant Treatment in Moderate-to-Severe Major Depressive Disorder: A Randomized Controlled Study"

_ijerph, 2022, doi:10.3390/ijerph20010091_

Round 1

Author Response

I appreciate the time and efforts by the editor and referees in reviewing this manuscript. We have addressed all issues indicated in the review report, and believed that the revised version can meet the journal publication requirements.

Introduction

  1. Introduction There is only little information on art therapy in the introduction. This could be elaborated, e.g. theoretical perspectives used in art therapy, proposed mechanisms of change., see for example Haeyen et al., 2020, Haeyen, et al., 2017, Bosgraaf, et al., 2022.

Authors’ Response : As you kindly suggested we have updated initial review of literature on art psychotherapy.

  • ‘Art psychotherapy is a form of psychotherapy that integrates creative processes and psychotherapeutic techniques. It uses art expression as a prime mode of communication; it conveys thoughts and feelings that cannot be explained verbally as an aesthetic product [32]. Images produced in art psychotherapy enhance the communication ability of inherent powers [31]. Art psychotherapy encourages individuals to explore personal meaning, recognize difficult emotions, integrate conflicting thoughts, and facilitate interpersonal relationships between individuals and art psychotherapists through the creative process [31, 33]’.
  • ‘The MDD Clinical Practice Guidelines (NICE 2014; SIGN, 2010) listed art psychotherapy as a nonpharmacological treatment strategy [36]’.

  1. In the paragraph on art therapy, I don’t understand the sentence “It is essential…mental illness” in the line of reasoning.

Authors’ Response : As you kindly suggested we have deleted the sentence.

  • ‘It is essential for clinicians to consider the fact that pharmacological agents have contributed to significantly lower quality of life and debilitating effects even when the pharmacotherapeutic strategies are clearly able to treat the major symptoms of a mental illness’

Materials and methods

  1. Sample size: the calculation is based on paired t-test, while for the overall statistical analyses repeated measures ANOVA were used. I would expect that the researchers based their sample size calculation on the repeated measures ANOVA.

Authors’ Response : We have newly added the result of sample size calculation on the repeated measures ANOVA.

  • ‘The sample size calculation was based on a repeated measures ANOVA (d= 0.5; α= .05), and a power of 95% was calculated using G*Power 3.1.9.2 for Windows [40]. Considering the dropout rate of 20%, the estimated total sample size required for the study was 38’.

  1. In order to interpret the results and its generalizability, it is necessary to receive more information on the way patients were selected in this specific psychiatric clinic. Can you give more information on the context/selection procedure? What was the procedure regarding referral of patients in this clinic to art therapy? In three years, only 45 patients were assessed eligible for this study.

Authors’ Response : As you kindly suggested we have included detailed information on the location of the clinic and the procedure regarding referral of patients.

  • ‘As shown in Figure 1, participants were recruited via a sampling frame of adult MDD patients from January 2014 to July 2015 (first phase) and from February 2017 to July 2017 (second phase) in a psychiatric outpatient clinic in the third largest district of Seoul, the Republic of Korea’.
  • ‘Self-referral methods were used to recruit participants. The poster was posted in the clinic waiting area, and the information brochure explained the clinical trial. Patients who wished to participate in the experiment were either contacted by art psychotherapists or clinicians for the initial screening arrangement’.

  1. What was the impact of using the criteria regarding ‘participated in less than three sessions of AT’ and ‘failure to maintain SSRIs dosages’? How many participants dropped out because of these criteria? What if you performed an intention to treat analysis? And thus don’t use these criteria. Why didn’t you use an intention to treat method? Please elaborate on this in the manuscript (perhaps as a limitation).

Authors’ Response : As you kindly suggested we have newly added

  • ‘To interpret the results of the experiment, we used per-protocol analysis. Six participants failed to maintain art psychotherapy and were excluded because of threats to internal variables. Specifically, the participants maintained a pharmacotherapy regime. However, they failed to maintain art psychotherapy, and all received a single art psychotherapy session. The exact reasons for the failure to maintain art psychotherapy could not be identified. A possible explanation for the failure to maintain treatment is that six continuous weekly in-person visits to the clinic may have caused treatment burden. This conjecture could be verified with an intention-to-treat (ITT) statistical analysis, which provides a realistic way to view a treatment regimen’.
  • ‘A per-protocol (PP) analysis was performed, including only participants who completed the experiment (Statistical Analyses)’.

  1. What were the reasons of the three participants that were excluded?

Authors’ Response : As you kindly suggested we have included a reason for exclusion.

‘Of the 45 participants, three were excluded. The reasons for declining participation were that two participants did not meet the inclusion criteria, and one participant declined to participate. A total of 42 participants were allocated to the experimental and control groups. Finally, 39 participants completed the experiment, while three dropped out. Two participants stopped clinic visits, and one participant withdrew consent’.

  1. Intervention: what was the therapeutic perspective used? What is meant with ‘depth of self-esteem’? this seems not in line with scientific literature on self-esteem.

Authors’ Response : As you kindly suggested we have deleted the sentence.

  • ‘of patients to evaluate the depth of self-esteem’

  1. Assessments: please mention also what kind of demographic information was collected.

Authors’ Response : As you kindly suggested we have added details on demographic information.

  • ‘Demographic information, including age, gender, and duration of illness, were collected’.

  1. Procedure: what is meant with ‘art psychotherapist collected implementation of the art psychotherapy?

Authors’ Response : As you kindly suggested we have revised the sentence.

‘The art psychotherapist conducted the BDI-II and collected the demographic information’.

Results

  1. In Table 1, moderate and severe depressive scores are mentioned. On what kind of measure is this based?

Authors’ Response : As you kindly suggested we have added details on depression severity

  • ‘Depression severities were determined by HDRS’.

  1. It is not clear to me whether the analysis on the secondary outcome were performed on only the experimental group or on both groups. Please mention the number of participants in Table 3.

Variables

Experimental Group

n

M(SD)

t

p

d

Pre

Post

HDRS1

Moderate

10

15.45(1.75)

11.91(2.59)

4.485***

0.001

1.35

Severe

9

25.09(4.04)

18.18(3.79)

6.820***

0.001

2.06

BDI-II2

Moderate

11

24.00(2.54)

20.70(5.29)

2.810*

0.020

0.89

Severe

8

35.64(7.75)

28.55(8.44)

5.722***

0.001

1.73

Authors’ Response : As you kindly suggested we have added the number of participants in Table 3.

  1. Please mention in the method section that the equality of covariance was a prerequisite for further analyzing the data.

Authors’ Response : As you kindly suggested we have newly added the sentence.

  1. The number of participants in Table 4 were very small and therefore I question the value of these results. I would recommend to delete this part.

Authors’ Response : As you kindly suggested we have deleted Table 4.

Discussion

  1. In the first paragraph, several equivalent studies were mentioned (18, 26-29, 47, 48). What were differences and/or similarities, especially regarding the intervention characteristics. The same applies to the second paragraph, (47, 48).

Authors’ Response : As you kindly suggested we have revised paragraphs.

  • ‘Furthermore, this finding is also in line with the results of a series of studies [18,26-29,47,48] that reported that add-on therapy or combined therapy was more effective in improving depressive symptoms than AD pharmacotherapy alone’.
  • ‘This finding is also in line with the results of a series of studies [18, 26-29,48,49] that reported that psychotherapy or combined therapy has clinically significant advantages over pharmacotherapy alone in treating depressive disorders. A meta-analysis of studies comparing psychotherapy with pill placebo [18] and a placebo-controlled, randomized trial that tested the relative efficacy of antidepressant medications (ADM) and cognitive therapy (CT) [48] showed that psychotherapy has an effect size comparable to that of ADM and outperforms placebo. A network meta-analysis that examined the relative effects of psychotherapies, ADs, and combined therapy for treating adult depression [26] showed that the combined treatment was more effective than psychotherapy or ADM alone in achieving response at the end of treatment. In addition, a study examining the relative efficacy of psychotherapy versus add-on therapy on individual depressive symptoms [27] showed that add-on therapy outperformed psychotherapy for treating some depressive symptoms except others. Furthermore, a review study [29] that evaluated the effects of combined therapy and ADM in reducing relapse and recurrence showed that concurrent therapy has a potential advantage in treating more severe depression. These trials have compared the relative effects of psychotherapies, ADM, and combined therapy with a representative sample of patients with mild-to-moderate [28] and moderate-to-severe depression [48,49]’.

  1. In my opinion, the part on remission rate is not convincing because of the small amount of participants that achieved remission. A difference of only one participant between the two groups could be a coincidence.

Authors’ Response : As you kindly suggested we have revised the discussion section.

  • ‘Although the remission rate is an important predictor of treatment outcome, the remission rate in the present study was very low to claim clinical meaning’.
  • ‘To interpret the results of the experiment, we used per-protocol analysis. Six participants failed to maintain art psychotherapy and were excluded because of threats to internal variables. Specifically, the participants maintained a pharmacotherapy regime. However, they failed to maintain art psychotherapy, and all received a single art psychotherapy session. The exact reasons for the failure to maintain art psychotherapy could not be identified. A possible explanation for the failure to maintain treatment is that six continuous weekly in-person visits to the clinic may have caused treatment burden. This conjecture could be verified with an intention-to-treat (ITT) statistical analysis, which provides a realistic way to view a treatment regimen’.

Reviewer 2 Report

Overall, I think this was a terrific paper. Great design resulting clear evidence of for the impact of six-week art therapy intervention as it impacts depression and PTSD. 

There is a need for language editing by a native English reader /writer including for the abstract and title, and I thought the discussion could illuminate more deeply what the authors imagine art therapy added to result in the reduction of depressive / ptsd symptoms after such a short intervention. There also should be some consideration of the impact of the specific art therapist or group as potential mediating factors, and condor the need to replicate the study with a comparative modality (verbal therapy? music therapy? etc) for stronger support of the findings.

All in all, well done!

Author Response

I appreciate the time and efforts by the editor and referees in reviewing this manuscript. We have addressed all issues indicated in the review report, and believed that the revised version can meet the journal publication requirements.

  1. There is a need for language editing by a native English reader /writer including for the abstract and title, and I thought the discussion could illuminate more deeply what the authors imagine art therapy added to result in the reduction of depressive / ptsd symptoms after such a short intervention. There also should be some consideration of the impact of the specific art therapist or group as potential mediating factors, and condor the need to replicate the study with a comparative modality (verbal therapy? music therapy? etc) for stronger support of the findings.

Authors’ Response : English language editing were made by a native speaker and intensive revision was conducted in order enhance the quality of the manuscript.
